# The drivers of mental health service utilisation among public primary healthcare workers in Negeri Sembilan

**Norhafizah Mohd Noor**[1,2]**, Ahmad Azuhairi Ariffin**[3]*****, Halimatus Sakdiah Minhat**[3]**, Lim Poh Ying**[3]**, Umi Adzlin Silim**[4]

**1** Faculty of Medicine and Health Sciences, University Putra Malaysia, Serdang, Selangor, Malaysia, **2** Setiu District Health Office, Permaisuri, Terengganu, Malaysia, **3** Department of Community Health, Faculty of Medicine and Health Sciences, University Putra Malaysia, Serdang, Selangor, Malaysia, **4** Department of Psychiatry, Serdang Hospital, Ministry of Health, Serdang, Malaysia

* zuhairifin@upm.edu.my

## Abstract

**Data Availability Statement:** Data cannot be shared publicly because our ethics committee does not allow it in order to protect the privacy of

### Background

Public primary healthcare workers (HCWs) face various psychosocial risks at workplace that can impact their mental health. However, little is known about their mental health service utilisation (MHSU). This study aimed to determine prevalence and predictors of MHSU among public primary HCWs in Negeri Sembilan, using Anderson Behavioural Model of Health Service Use.

### Methods

A cross-sectional study was conducted from December 2022 to April 2023, using a valid and reliable self-administered six sections questionnaire consisting of; (i) sociodemographic, (ii) work-related factors, (iii) MHSU, (iv) perception of stigmatisation by others, (v) enabling factors, and (vi) need factors. Respondents were selected through proportionate stratified random sampling based on job categories. Multiple Logistic Regression using SPSS version 26 was used to determine the predictors of MHSU.

### Results

A total of 294 respondents participated in this study, with a response rate of 83.5%. The 12-months MHSU prevalence was 45.6%. Mental health services were predominantly utilised for screening (96.3%) and treatment purposes (28.4%), primarily accessed through health clinics (85.1%), and interaction with paramedics (44.0%) and medical officers (38.8%). Significant drivers predicting MHSU were B40 household income (aOR = 3.426, 95% CI: 1.588, 7.393, p-value = 0.002) and M40 household income (aOR = 3.781, 95% CI: 1.916, 7.460, p-value<0.001), low supervisor support (aOR = 2.302, 95% CI: 1.206, 4.392, p-value = 0.011), received mental health training (aOR = 2.058, 95% CI: 1.221, 3.469, p-value = 0.007) and high co-worker support (aOR = 1.701, 95% CI: 1.034, 2.798, p-value = 0.036).

participants. The data are securely stored in a server at Universiti Putra Malaysia (UPM) and are accessible only to the research team. Data requests can be directed to the corresponding author, Associate Prof. Dr. Ahmad Azuhairi Ariffin (email: zuhairifin@upm.edu.my), and will be subject to approval by both the Medical Research and Ethics Committee of the Ministry of Health (https://nmrr. gov.my/) and the Ethics Committee for Research Involving Human Subjects at UPM (https://tncpi. upm.edu.my/services/research_ethics_evaluation/ jkeupm_ethic_committee_for_research_involving_ human_subject-39931).

**Funding:** This study was funded by the Geran Berfocus JKK (GP-F/JKK), Award Number: 6302007-14001. The recipient of this grant was Lim Poh Ying, PhD in Biostatistics, University of Nottingham, UK. No additional external funding was received for this study.

**Competing interests:** The authors have declared that no competing interests exist.

## Conclusion

Almost half of respondents used mental health services, predicted by lower and middle household income, received mental health training and high co-worker support. Conversely, MHSU was also predicted by low supervisor support. To sustain high levels of MHSU, it is essential to implement regular mental health training targeted HCWs with lower to middle household incomes and those experiencing low supervisor support, while simultaneously enhancing co-worker support and screening program for early detection of mental health problems.

## Introduction

Public primary healthcare workers (HCWs) play an important role in integrating care, prevention and management encompassing assessment, treatment and rehabilitation services [1]. This is in line with the Sustainable Developmental Goals, with the target of 3.8 which aim to achieve universal health coverage by providing financial risk protection, access to quality healthcare services, essential medication and vaccine for all [2]. As frontliners in community services, they are no strangers to challenges, experiencing various psychosocial risks at their workplace that could have impacts on their mental health [3]. Excessive workloads, limited resources, low job support, interpersonal relations conflicts and poor work environments led to increased risks of mental health problems among public primary HCWs [3]. Whether they choose to confront these challenges or decide to quit, the possible impacts are undeniable.

In spite of the COVID-19 pandemic that led to a notable increase in workplace psychosocial risk and a high prevalence of mental health problems, the utilisation of mental health services remains low [4–9]. The systematic review of mental health problems among HCWs managing the COVID-19 pandemic revealed varying prevalence rates: anxiety had the highest rate at 67.6%, followed by stress at 62.9% and depression at 55.9%. While the lowest rates were observed for depression at 12.1%, stress at 29.8%, and anxiety at 24.1% [10]. In Malaysia, the prevalence of depression, anxiety, and post-traumatic stress disorder (PTSD) among HCWs during COVID-19 pandemic was 22.7% for depression [11], 29.7% for anxiety [12] and 25% for PTSD [13] in various study locations and by using assorted tools. Besides, the national study on psychological effect of COVID-19 pandemic among Malaysian HCWs in private and public sectors demonstrated that assistant environment health officers were reported to have the highest percentage (26.3%) of more likely to have severe mental health disorders [14]. On top of their regular duties, the assistant environment health officers have prolonged working hours, heavily involved in COVID-19 contact tracing, mass sampling, handling data and decontamination procedures [15].

The global prevalence of mental health service utilisation (MHSU) and professional help-seeking among HCWs were 10.9% to 12.7% in China, 13% in India, 13% and 18% in Australia, 19.1% in USA and 20% in Canada [4–9]. The main reasons for HCWs not utilising mental health services include concern about confidentiality and privacy, stigma, lack of time to seek help, perception of mental health as low priority, problem not severe enough, impact on career progression, denial, lack of illness acceptance, limited access to the services, negative evaluation of services and shortage of mental health professionals [5, 9, 16–21]. In addition, previous studies highlight that HCWs, primarily doctors will seek professional help when their mental health symptoms become severe and cause significant occupational impairment [22, 23].

The impact of low MHSU among HCWs can be far reaching. Lower utilisation of mental health services leads to delays in diagnosis and treatment, progression to severe mental disorders and increased risk of suicide [24]. It can also lead to occupational impairment, poor judgment, and decision-making, which could pose a greater risk to patient safety and increase the likelihood of medico-legal issues, increased absenteeism and heightened job resignation [20, 24–27].

Recognising the profound implications of untreated mental health problems among HCWs, this study is critical to bridge the gaps through comprehensive research. Globally, majority of research primarily focuses on treatment and seeking professional help for mental health problems; screening is not addressed [4–9]. Thus, understanding factors associated with MHSU from the screening utilisation is important for early recognition of mental health problems and facilitate treatment utilisation among HCWs [28]. In the context of Malaysia, there is a notable research gap concerning the MHSU among HCWs. Despite the importance of understanding the factors influencing service utilisation, there are no published studies examining this phenomenon within Malaysian HCWs. It is anticipated that this research will yield valuable insights for the formulation of targeted interventions, enhancing support systems and planning for mental health service for HCWs. This study represents a pioneering effort within the context of Malaysian HCWs and providing novel insights into the MHSU concerning comprehensive services including screening, treatment and rehabilitation. Therefore, this study aims to determine the prevalence and predictors of MHSU among public primary HCWs in Negeri Sembilan using the Anderson Behavioural Model of Health Service Use (ABMHSU).

## Materials and methods

One of the most widely used models to comprehend MHSU is by utilising Anderson Behavioural Model of Health Service Use (ABMHSU). This model offers a robust framework that can be used to ascertain both the conditions in facilitating or identifying the barriers to the utilisation of services by HCWs. Three sets of factors influenced this model including (i) predisposing factors, (ii) enabling factors, and (iii) the need factors [29]. In the present study, we examined the factors associated with MHSU using ABMHSU, including:

(i) Predisposing factors encompass individual characteristics that influence health service utilisation include age, gender, prior history of mental illness, place of duty, job category, working hours, overnight work, mental health training, and perception of stigma by others.

(ii) Enabling factors are resources that may facilitate access to healthcare include supervisor support, co-worker support, awareness of mental health services, and household income.

(iii) The need factors represent potential needs for health service use include self-rated health status, depression, anxiety, and stress.

### Study design and location

This is a cross-sectional study, conducted among public primary HCWs in the State of Negeri Sembilan. Negeri Sembilan is situated in the western region of Peninsular Malaysia and covers an area of 6,658 km$^2$ with a population of approximately 1.13 million [30]. The location was chosen in accordance to the National Health Morbidity Survey 2019, Negeri Sembilan was found to be the top three states having the highest prevalence of depression among adult population [31]. Negeri Sembilan State Health Office oversees seven district health offices and 51 health clinics. One district health office was designated for this pilot study, while the other remaining district health offices and clinics participated in the study.

## Inclusion and exclusion criteria

Inclusion Criteria:

(1) HCWs working in district health offices and health clinics in Negeri Sembilan

(2) Four job categories including (i) doctors, (ii) nurses, (iii) assistant medical officers, (iv) assistant environment health officers and environment health officers.

Exclusion Criteria:

(1) HCWs with working experience of fewer than three months,

(2) those on maternity leave during the study period and

(3) HCWs who refuse to participate in the study.

## Sample size

The sample size was calculated using two independent proportion formula by Lwanga and Lemeshow 1991 [32]. The reason of using the largest sample size possible and with the most feasible sample size in conducting the research as it is within the achievable time frame. The sample size was chosen based on a study by Tian et al. (2021). P1 is the proportion of respondents with depressive symptoms who have used mental health services (20.2%), whereas P2 is those who did not have depressive symptoms but have used mental health services (8.7%) [4]. Based on the calculation, sample size estimation (n) is 146. The total sample size required is 352 after the adjustment of comparison of two groups and 20% non-response rate.

## Sampling method

All six district health offices were included in the study, with the Port Dickson District Health Office reserved for the pilot study. Sampling frames were obtained from Negeri Sembilan State Health Department and all the district health offices. The required samples were selected using proportionate stratified random sampling with job categories served as strata. The strata consist of four job categories which are (i) doctor, (ii) nurse, (iii) assistant medical officer, and (iv) assistant environment health officer & environment health officer. This method was chosen due to the heterogeneity of the study sample. The number of samples required in each stratum is derived by calculating the entire sample size divided by the total population in sampling frame times with the total numbers of HCW in each stratum. The selection of eligible respondents from each stratum was done using simple random sampling. Identification of the data number was assigned to each respondent in the sampling population and random numbers were generated using Microsoft Excel.

## Study instruments and quality control

A Malay Language (Bahasa Malaysia) self-administered paper-based questionnaire was used to reduce the non-response rates, as demonstrated in a previous study [33]. The questionnaires consist of six sections. The first and second sections include self-constructed questions on sociodemographic factors like age, gender, history of mental illness, and a work-related section consisting of place of work, job category, working hours, overnight work, and mental health training. The third section consists of questions on MHSU adapted from Wang et al. (2007) [34]. Participants were asked if they had used mental health services for screening, counselling, treatment, and rehabilitation in the past 12 months. If yes, they were asked about the purpose,

frequency, type of service, and professional met. If no, they were asked about the main reasons for non-use. Forth section measures "Stigma by Others", using "Perception of Stigmatisation by Others for Seeking Help (PSOSH)" tool, developed by Vogel et. al (2009), translated and validated into Malay version with a good Cronbach's alpha value of 0.84 [35]. PSOSH scale consists of 5 questions, measured on 5-point Likert scale and total score ranges from 5 to 25. The fifth section measures the enabling factors, such as household income per month, awareness of mental health services, co-workers, and supervisor support. The co-worker and supervisor support were adapted from Malay Language version Job Content Questionnaire (JCQ) by Amin et al. (2015). It has good internal consistency with Cronbach alpha value of 0.83 [36]. Supervisor support and co-worker support consist of four questions in each domain, measured on 4-point Likert scale with total scores ranging from 4 to 16. The sixth section comprised need factors including self-rated health and mental health status. Self-rated health is a single question, measured using 5-point Likert scale ranging from 1–5. The question asked respondents to rate their general health in the past two weeks. The scales 1–2 were categorised as good and 3–5 were categorised as poor self-rated health status [37]. Mental health status was measured using Malay Depression, Anxiety and Stress 21-item (DASS-21) questionnaire. It consisted of 21 questions, each question using 4- point Likert scale (from 0 to 3). A higher score indicates higher level of the respective subscale's symptoms. This tool was adopted and validated by Musa et al. (2007) with acceptable to good Cronbach's alphas values 0.84, 0.74, and 0.79 for depression, anxiety, and stress, respectively [38]. Scores of more than 5 were interpreted as "yes for depression", more than 4 as "yes for anxiety", and more than 7 as "yes for stress", respectively, based on the items of each subdomain [39].

Before the actual study, the face and content validity were evaluated by four public health physicians and a consultant psychiatrist. The details of the responses to the questionnaire were thoroughly evaluated and revised accordingly. All 38 appropriate items were retained based on their content validity index. For the reliability test, a pilot study was performed on 30 HCWs at the Port Dickson Health Office in Negeri Sembilan, who were not sampled in the actual study. The overall result demonstrated good to excellent internal consistency with Cronbach alpha's values 0.81 for stress, 0.78 for anxiety, 0.86 for depression, 0.92 for supervisor support, 0.89 for co-worker's support and 0.93 for PSOSH scale. The MHSU questions also demonstrated an excellent test-retest reliability with Cohen's kappa = 1. It is imperative to verify that the DASS-21, PSOSH scale, and JCQ scale were reliable instruments for evaluating depression, anxiety, and stress, stigma, supervisor and coworker support respectively. This validation helps to account for cultural, demographic, and contextual disparities within the study population that may influence the interpretation. Furthermore, MHSU questions signifies perfect test-retest reliability assessment and confirm its stability and consistency in measuring the utilisation of mental health services over time.

## Data collection method

Data was collected from eligible HCWs from December 2022 to April 2023. Eligible HCWs were approached individually at their workplace, with their approved consent, and personal information sheet before eventually answering the questionnaire. The questionnaires were completed in 20 minutes, and the researcher thoroughly checked for missing information upon submission.

## Data analysis

Data was analysed using IBM Statistical Package for Social Sciences version 26. Descriptive statistics were presented in frequency and percentage for the categorical data, mean and standard

deviation or median with inter-quartile range for continuous data. This study utilised Multiple Logistic Regression (MLR) to determine predictors of MHSU. Variables that demonstrated a p-value of less than 0.25 in simple logistic regression were selected for inclusion in the MLR model [40]. For determining statistical significance in the MLR, a p-value threshold of 0.05 and a 95% confidence interval (CI) were applied to all tests.

### Ethical consideration

The study received ethical approval from the National Medical Research Register and the Medical Research and Ethics Committee of the Ministry of Health, Malaysia with approval number NMRR-ID-22-02258-CTA (IIR). Following this approval, formal notifications were sent to Negeri Sembilan State Health Office, and all participating District Health Offices. Participants were provided with an information sheet and were required to sign a written informed consent form prior to participation. Throughout the study, the researcher adhered to the Declaration of Helsinki and Malaysian Good Clinical Practice Guidelines.

## Results

### Prevalence of mental health service utilisation (MHSU)

A total of 294 out of 352 respondents completed the self-administered questionnaire, representing an 83.5% response rate. The 12-month prevalence of MHSU was 45.6%. Among those utilising mental health service, majority used it once (91.0%), for screening (96.3%) and treatment purposes (28.4%), primarily accessed through health clinics (85.1%), and involved interaction with paramedics (44.0%) and medical officers (38.8%).

Conversely, 54.4% of respondents did not use mental health services. The main reasons cited were no mental health problems (76.3%), problems not severe enough (26.9%), not enough time (7.5%), privacy concerns (5.6%), and others (see Table 1).

### Characteristics of respondents

Table 2 shows the characteristics of respondents based on predisposing, enabling and need factors. For predisposing factors, the median age of respondents was 35 (IQR = 9), and the median years of service were 11 years (IQR = 9). Most respondents were female (73.5%), married (82.7%) and had no history of mental illness (94.9%). For the job characteristics, 50.3% of respondents were nurses, 23.5% doctors, 16.3% assistant medical officers, and 9.9% assistant environmental health officers and environmental health officers. Most of respondent worked at health clinics (89.5%) and did not receive mental health training in the past 12 months (66.0%). The median day of overnight work in previous week was zero (IQR = 0), median of average working hours was 45 hours per week (IQR = 5) and median score for stigmatisation by others for seeking help was 9 (IQR = 8). For the enabling factors, most of them are from M40 household income (54.4%), majority aware of available mental health services (99.7%) and most of respondents had received high supervisor and high co-worker support with 81.3% and 57.5%, respectively. In this study, household income is defined as the respondents' monthly gross household income. Household incomes were categorised into three groups based on gross household income in Negeri Sembilan: less than RM 4,210 for B40, RM 4,210 to RM 9,299 for M40, and RM 9,300 or more for T20 [41]. As for the need factors, most respondents were in a good self-rated health status (82.7%). In relation to the mental health status using DASS-21 tool, the highest prevalence was anxiety (15%), followed by depression (11.6%) and stress (10.9%).

**Table 1. Characteristics and prevalence of MHSU (N = 294).**

| Characteristics | Frequency (n) | Percentage (%) |
|---|---|---|
| **MHSU past 12 months** | | |
| No | 160 | 54.4% |
| Yes | 134 | 45.6% |
| **Frequency of MHSU in the past 12 months (n = 134)** | | |
| Once | 122 | 91.0% |
| Twice | 10 | 7.5% |
| >Twice | 2 | 1.5% |
| **Reasons for MHSU** [a] | | |
| Screening | 129 | 96.3% |
| Treatment | 38 | 28.4% |
| Rehabilitation | 1 | 0.7% |
| **Type of facilities utilised** [a] | | |
| Health Clinic | 114 | 85.1% |
| Occupational Health Clinic | 15 | 11.2% |
| Counselling Unit | 3 | 2.2% |
| Others | 3 | 2.2% |
| Public Hospital | 2 | 1.5% |
| Private Clinic | 2 | 1.5% |
| Psychiatric Specialist Clinic | 2 | 1.5% |
| **Type of professional met** [a] | | |
| Paramedic (Nurse/Assistance Medical Officer) | 59 | 44.0% |
| Medical Officer | 52 | 38.8% |
| Occupational Health Doctor | 12 | 9.0% |
| Counsellor | 7 | 5.2% |
| Clinical Psychologist | 6 | 4.5% |
| Occupational therapist | 5 | 3.7% |
| Psychiatrist | 1 | 0.7% |
| Social worker officer | 1 | 0.7% |
| **Reasons not utilised MHS** [a] | | |
| No mental health problems | 122 | 76.3% |
| Problems not severe enough | 43 | 26.9% |
| Not enough time | 12 | 7.5% |
| Concern about privacy | 9 | 5.6% |
| Concern about career implication | 6 | 3.8% |
| Stigma | 5 | 3.1% |
| Difficulty to get mental health services | 4 | 2.5% |
| Others | 8 | 5.0% |

[a] A respondent can choose more than one option

## Predictors of MHSU

Seven variables with p-value <0.25 from simple logistic regression (SLR) were tested in multiple logistic regression (MLR) namely, mental health training (p-value = 0.020), job category (p-value = 0.073), household income (p-value = 0.001), average working hours (p-value = 0.171), supervisor support (p-value = 0.075), co-worker support (p-value = 0.034) and self-rated health status (p-value = 0.246). It has been suggested that the variables with p-value<0.25 in the

**Table 2. Characteristics of respondents according to predisposing, enabling and need factors (N = 294).**

| Characteristics | Frequency (n) | Percentage (%) |
|---|---|---|
| | Median (IQR) | |
| **Predisposing Factors** | | |
| **Age (years)** | 35 (9) | |
| **Years of services** | 11 (9) | |
| **Gender** | | |
| Male | 78 | 26.5% |
| Female | 216 | 73.5% |
| **Marital status** | | |
| Single | 42 | 14.3% |
| Married | 243 | 82.7% |
| Divorce and Widow | 9 | 3.0% |
| **Prior history of mental illness** | | |
| No | 279 | 94.9% |
| Yes | 11 | 3.7% |
| Not sure | 4 | 1.4% |
| **Job category** | | |
| Assistant environment health officers & environmental health officers | 29 | 9.9% |
| Assistant medical officer | 48 | 16.3% |
| Doctor | 69 | 23.5% |
| Nurse | 148 | 50.3% |
| **Place of duty** | | |
| District Health Office | 31 | 10.5% |
| Health Clinic | 263 | 89.5% |
| **Days of overnight work in past week** | 0 (0) | |
| **Working hours per week in past week** | 45 (5) | |
| **Mental health training** | | |
| No | 194 | 66.0% |
| Yes | 100 | 34.0% |
| **Perception stigmatisation by others** | 9 (8) | |
| **Enabling Factors** | | |
| **Household income per month** | | |
| < RM4210 (B40) | 72 | 24.5% |
| RM4210-9299 (M40) | 160 | 54.4% |
| ≥RM9300 (T20) | 62 | 21.1% |
| **Awareness of available mental health services (MHS)** | | |
| No | 1 | 0.3% |
| Yes | 293 | 99.7% |
| **Supervisor support** | | |
| Low support (<12) | 55 | 18.7% |
| High support (≥12) | 239 | 81.3% |
| **Co-worker support** | | |
| Low support (<13) | 125 | 42.5% |
| High support (≥13) | 169 | 57.5% |
| **Need Factors** | | |
| **Self-rated health status** | | |
| Good | 243 | 82.7% |

*(Continued)*

**Table 2.** (Continued)

| Characteristics | Frequency (n) | Percentage (%) |
|---|---|---|
| | | Median (IQR) |
| Poor | 51 | 17.3% |
| **Stress** | | |
| No | 262 | 89.1% |
| Yes | 32 | 10.9% |
| **Anxiety** | | |
| No | 250 | 85.0% |
| Yes | 44 | 15.0% |
| **Depression** | | |
| No | 260 | 88.4% |
| Yes | 34 | 11.6% |

univariate analysis and clinically important variables should be included in the multivariate analysis [40]. This is due to the weak association with the outcome in univariate analysis that can contribute significantly when they are combined.

Table 3 shows four significant predictors for MHSU in the final model, using forward LR variable selection method. The model assumption was checked. The overall classification percentage was 63.3%. No multicollinearity between the variables. The model had significant model fit (Hosmer and Lemeshow test, p-value = 0.373) with Cox and Snell R squared and Nagelkerke R squared were 10.3% and 13.8% respectively. Based on the Receiver Operating

**Table 3. Predictors of MHSU using multiple logistic regression analysis (N = 294).**

| Variable | Adjusted B | SE | Wald | p-value | aOR | 95% CI of aOR | |
|---|---|---|---|---|---|---|---|
| | | | | | | Lower | Upper |
| **Intercept** | -1.938 | 0.375 | | | | | |
| **Household income per month** | | | | | | | |
| < RM4210 (B40) | 1.231 | 0.392 | 9.845 | 0.002* | 3.426 | 1.588 | 7.393 |
| RM4210-9299 (M40) | 1.330 | 0.347 | 14.711 | <0.001* | 3.781 | 1.916 | 7.460 |
| ≥RM9300 (T20) | Ref | | | | | | |
| **Mental health training** | | | | | | | |
| No | Ref | | | | | | |
| Yes | 0.722 | 0.266 | 7.347 | 0.007* | 2.058 | 1.221 | 3.469 |
| **Supervisor support** | | | | | | | |
| Low support | 0.834 | 0.330 | 6.397 | 0.011* | 2.302 | 1.206 | 4.392 |
| High support | Ref | | | | | | |
| **Co-worker support** | | | | | | | |
| Low support | Ref | | | | | | |
| High support | 0.531 | 0.254 | 4.375 | 0.036* | 1.701 | 1.034 | 2.798 |

Note:

*Significant at p-value <0.05, aOR is adjusted odd ratio, CI is confident interval

Forward LR was applied, no multicollinearity and no interaction terms

Hosmer and Lemeshow test (p-value = 0.373), classification table (overall percentage: 63.3%)

Cox and Snell R squared (0.103), Nagelkerke R squared (0.138), ROC = 0.687

Characteristic (ROC) curve, the model was significantly discriminated in 68.7% of cases (ROC = 0.687, p-value<0.001).

The result showed that respondents with household income in B40 and M40 groups had 3.426 times and 3.781 times more likely to utilise mental health services as compared to T20 groups, respectively (aOR = 3.426, 95% CI: 1.588, 7.393, p-value = 0.002; aOR = 3.781, 95% CI: 1.916, 7.460, p-value<0.001). Respondents with low supervisor support were 2.302 times more likely to utilise mental health services as compared to those who received high supervisor support (aOR = 2.302, 95% CI: 1.206, 4.392, p-value = 0.011). Respondents who received mental health training were 2.058 times more likely to utilise mental health services as compared to those who did not receive mental health training (aOR = 2.058, 95% CI: 1.221, 3.469, p-value = 0.007). Respondents with high co-worker support were 1.701 times more likely to utilise mental health services as compared to those who received low co-worker support (aOR = 1.701, 95% CI: 1.034, 2.798, p-value = 0.03).

## Discussion

### Mental health service utilisation (MHSU)

To our knowledge, this study represents the inaugural attempt in Malaysia to determine the prevalence and predictors of MHSU among public primary HCWs. The current study found that 45.6% of public primary HCWs in Negeri Sembilan had MHSU in the past 12 months. The prevalence is notably high as compared to lifetime MHSU by nurses in China (10.9%) and MHSU for treatment among frontline HCWs during COVID-19 pandemic in United States (19.1%) [4, 8]. The high prevalence of service utilisation in our study was attributed to screening purposes, which accounted for 96.3% of cases among those who utilised mental health services. Studies in China and United States did not include MHSU for screening. Notwithstanding this, our study observed a relatively high prevalence of MHSU for treatment at 28.5%, in comparison to prior studies [4, 8]. Mental health screening is an essential component of preventive healthcare as it enables the prompt detection of mental health issues. It facilitates treatment access, safeguards confidentiality, and encourages the normalisation of mental health discussions, thereby reducing stigma. Furthermore, the implementation of the KOSPEN WOW (Wellness of Workers) by Malaysian Ministry of Health since 2016 had increased accessibility and encouraged MHSU among HCWs [42]. This program provides mental health training, voluntary mental health screenings, and a connection to active intervention [43]. This is consistent with the findings of a systematic review and meta-analysis on mental health screening in the workplace, which concluded that screening followed by appropriate treatment interventions had a more beneficial impact on the mental health of employees [28].

Moreover, the COVID-19 pandemic acted as a catalyst for a significant surge in awareness pertaining to mental health, resulting in heightened levels of self-awareness and advocacy. These developments in turn encouraged HCWs to participate in screening activities and seek treatment. Significant efforts have been made to enhance mental health services in Malaysia during the COVID-19 pandemic, such as digitalising mental health screening, creating of national helpline service "Talian HEAL" and establishment of National Centre of Mental Health Excellence to coordinate mental health services across public, private, and non-governmental organisations [44].

### Mental health training

In the current study, mental health training emerged as the only significant predictor for predisposing domain from ABMHSU. This finding is aligned with a cross-sectional study among public health workers in China demonstrated that psychological training enhances the

likelihood of seeking professional help during the COVID-19 pandemic. The beneficial impact of mental health training enhances knowledge, and awareness, fostering positive attitudes and reducing the stigma towards MHSU. This is explained by a mixed-method study among doctors in Australia, which emphasized that the recognition and awareness of symptoms of mental health problems are required to prompt doctors to seek help [45]. Besides, the positive view about mental health has encouraged NHS doctors to utilise mental health services [23]. Similarly, a qualitative study by Spiers et al. (2017) mentioned how openness and low stigma could encourage MHSU. The willingness to discuss mental health issues and how they benefited from seeking help encourages others with similar issues to seek professional help [22]. Therefore, it is recommended that regular mental health training integrated with screening programs presents a golden opportunity to enhance MHSU and optimise mental health outcomes among HCWs.

## Household income

Three significant predictors identified in the enabler domain of ABMHSU encompassing the M40 and B40 household income group, high co-worker support and low supervisor support. A strong predictor of MHSU is the B40 and M40 household income. There is a growing chance that people from lower and middle household incomes use mental health services more frequently because of financial stress, lead to mental health problems and raises the need for such services. A systematic review reveals that lower household-income employees are more likely to have poor mental health outcomes such as depression, anxiety, stress and psychological distress [46]. A nationwide survey of Malaysia's adult population supports this statement, finding that the proportion of depressed people in the B40 (2.7%) and M40 (1.7%) groups was higher compared to T20 groups (0.5%) [47]. In addition, the majority of respondents (85.1%) reported that they had utilised mental health services at public health clinics. Public mental health services in Malaysia are provided free of charge to all HCWs by government-affiliated entities. Additionally, they are familiar with the system and have easy access to internal services. This is supported by a cross-sectional study by 7846 Australian HCWs showing that nurses and allied health workers are more likely to utilise internal mental health services [48]. However, the findings contrast with the mixed method study findings among pharmacists in the United States which found that low income is a barrier to utilising mental health services as the informant explained that adequate finances are important for sustainable follow-up and treatment [19].

## Supervisor support

Another factor that predicts MHSU in public primary HCWs is low supervisor support. This can be explained by low supervisor support in terms of lack of emotional and practical support as well as poor relationship with supervisor that can lead to the increased risk of poor mental health outcomes. The findings of a systematic review and meta-analysis of 22 studies demonstrated that receiving low supervisor support was associated with 1.45 and 1.16 times higher risk of suicidal ideation and suicide, respectively [49]. Additionally, interpersonal conflicts at work, such as violence and workplace bullying, have been shown to elevate the risk of stress, anxiety, and depression [50]. Furthermore, an effort-reward imbalance, where high effort at work is coupled with low appreciation, has been linked to an increased risk of depression [51]. These factors could indirectly contribute to an increase in MHSU. Besides, the Job Demand-Resource Model elucidates the role of supervisor support in the workplace. Specifically, it indicates that low supervisor support serves as a risk factor, exacerbating the negative effects of job demands and increased workplace stressors [52].

However, this finding differs from a qualitative study among NHS doctors, which discovered that supportive supervisors can encourage help-seeking behaviour due to their elevated status within the hierarchy, which allows them to successfully encourage junior doctors to use mental health services [23]. In this study, the low supervisor support acts as a stressor, thereby influencing MHSU differently. The distant relationship and poor supervisor support would impede open conversation and result in reluctant to interact with supervisor [53]. As a result, HCWs opt to communicate with their colleagues, who link them to mental health services or they directly used other preferable mental health [45, 54, 55]

## Co-worker support

Another significant predictor for MHSU is high co-worker support. The result aligns with the qualitative studies among doctors which explained that informal conversations between colleagues are able to facilitate access to mental health services [54, 55]. Most doctors prefer informal consultation and are likely to contact friends, colleagues and family who are also doctors if they have mental health problems or difficulties [55]. Another qualitative study among doctors in Norway mentioned that co-worker support provides psychological safety, trust, time flexibility, informal talk and raising awareness to seek help and MHSU [54]. Furthermore, most HCWs are likely to turn to their co-workers when encountering difficulties [56, 57]. Considering these findings, it is critical to develop robust peer support networks to promote informal support mechanisms as essential elements of MHSU and to create a supportive work environment.

## Strength and limitation

By investigated MHSU among HCWs in Malaysia, this study able to bridge the knowledge gap and laying the groundwork for future research. Furthermore, this encompasses the role of screening in MHSU, which highlights how screening serves as a gateway to treatment utilisation, especially when integrated with training. Besides, it also utilised locally developed validated and reliable questionnaires for the local study population to ensure high-quality data collection. The research had been strengthened by employing ABMHSU as the primary model. Information gathered from this research can also be applied to develop ABMHSU model-based intervention for MHSU.

However, some limitations of this study should be highlighted. The cross-sectional study restricts the ability to establish a causal relationship between factors associated and MHSU as both are assessed at one point of time. Other than that, the information was collected through retrospective self-reports of MHSU, which potentially resulted in recall bias and social desirability. Respondents may give inaccurate recalls or not declare their past MHSU.

## Recommendation for future research

This study lays the groundwork for future research that could expand the scope of studies by conducting prospective cohort study to establish temporal relationship. In addition, the data was gathered through self-reporting MHSU, should be validated with medical record or screening surveillance to prevent recall bias. As this study only focus on individual characteristic in ABMHSU, future study should focus on other factors such as environment, healthcare providers and healthcare system [29]. Other mental health problems encountered by HCWs such as adjustment disorder, work related stress, burnout, grief, post-traumatic stress disorder should be examined as well.

## Conclusion

Approximately half of the total number of respondents used mental health services. This was predicted by driving factors such as receiving mental health training, having high co-worker support, and lower and middle household income. Conversely, MHSU was also predicted by low supervisor support. In order to sustain high levels of MHSU, it is imperative to provide regular mental health training specifically for HCWs from lower to middle household income and those with low supervisor support. Additionally, enhancing coworker support and implementing robust screening programs for the early detection of mental health problems are essential.

## Acknowledgments

We express our gratitude to the Director-General of Health Malaysia for authorising us to publish this article. Additionally, we would like to extend our sincere appreciation to the Negeri Sembilan State Health Director, all District Health Officers, the Medical Officer in Charge, Matron, Chief of Assistant Medical Officers, and Environment Health Officers in allowing and providing us the permission to conduct this study. We also like to extend sincere gratitude to all the participants for their contribution and efforts in conducting this study. Lastly, our sincere appreciation to all UPM lecturers and DrPH class of 2021–2024 for their valuable advise and remarkable support.

## Author Contributions

**Conceptualization:** Norhafizah Mohd Noor, Ahmad Azuhairi Ariffin, Halimatus Sakdiah Minhat.

**Data curation:** Norhafizah Mohd Noor.

**Formal analysis:** Norhafizah Mohd Noor, Lim Poh Ying.

**Funding acquisition:** Norhafizah Mohd Noor, Ahmad Azuhairi Ariffin, Halimatus Sakdiah Minhat.

**Investigation:** Norhafizah Mohd Noor.

**Methodology:** Norhafizah Mohd Noor, Ahmad Azuhairi Ariffin, Halimatus Sakdiah Minhat.

**Project administration:** Norhafizah Mohd Noor, Ahmad Azuhairi Ariffin.

**Resources:** Norhafizah Mohd Noor, Ahmad Azuhairi Ariffin, Halimatus Sakdiah Minhat, Lim Poh Ying.

**Supervision:** Ahmad Azuhairi Ariffin, Halimatus Sakdiah Minhat, Lim Poh Ying, Umi Adzlin Silim.

**Validation:** Norhafizah Mohd Noor, Ahmad Azuhairi Ariffin, Halimatus Sakdiah Minhat, Lim Poh Ying, Umi Adzlin Silim.

**Visualization:** Norhafizah Mohd Noor.

**Writing – original draft:** Norhafizah Mohd Noor.

**Writing – review & editing:** Norhafizah Mohd Noor, Ahmad Azuhairi Ariffin, Halimatus Sakdiah Minhat, Lim Poh Ying, Umi Adzlin Silim.

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
