## [Decision Letter · Decision Letter 0]

14 Jun 2024

PONE-D-24-20993The drivers of mental health service utilisation among public primary healthcare workers in Negeri SembilanPLOS ONE

Dear Dr. Mohd Noor,

Thank you for submitting your manuscript to PLOS ONE. After careful consideration, we feel that it has merit but does not fully meet PLOS ONE’s publication criteria as it currently stands. Therefore, we invite you to submit a revised version of the manuscript that addresses the points raised during the review process.

We look forward to receiving your revised manuscript.

Kind regards,

Tariq Jamal Siddiqi

Academic Editor

PLOS ONE

Journal Requirements:

"This study was supported by the Geran Putra Berfocus JKK (GP-F/JKK), University Putra Malaysia"

Reviewers' comments:

Reviewer's Responses to Questions

**Comments to the Author**

1. Is the manuscript technically sound, and do the data support the conclusions?

Reviewer #1: Yes

Reviewer #2: Yes

2. Has the statistical analysis been performed appropriately and rigorously? 

Reviewer #1: Yes

Reviewer #2: Yes

3. Have the authors made all data underlying the findings in their manuscript fully available?

Reviewer #1: Yes

Reviewer #2: Yes

4. Is the manuscript presented in an intelligible fashion and written in standard English?

Reviewer #1: Yes

Reviewer #2: Yes

5. Review Comments to the Author

Reviewer #1: In the introduction, it is unclear what the authors mean by target 3.8 in sustainable developmental goals. It would be helpful if the authors could explicitly state what the specific goal or target is that they are referring to in here

The authors should further elaborate the rationale for this study

The introduction of this study is very lengthy. Authors can benefit by reducing and excluding unnecessary or redundant information that they have discussed in the intro, - a lot of these points can be added to the discussion or be summarized and discussed in a concise manner

Was an external review board certification/permission required for this study?

Confidence interval abbreviation can be added in the data analysis part

The abbreviated and full form of MHSU should be provided

Even though the authors have mentioned the income categories in the table they need to provide the definitions for the income categories B40, M40 and T20 within the results text to ensure clarity for the readers

The statement “This is consistent with the findings of a systematic review and meta-analysis on mental health screening...” is very confusing. It is not clear what specific results or points the authors are referring to. The sentence should be restructured to clearly indicate which findings are being referenced in here.

This paper can also benefit from a 'brief' paragraph on future directions for investigators and physicians, and how future studies could explore and use more rigorous analysis to further improve the findings/observations of this study.

Reviewer #2: In their study, " The drivers of mental health service utilization among public primary healthcare workers in Negeri Sembilan," Noor et al. investigated the prevalence and predictors of MHSU among HCWs. Their findings revealed that low supervisor support, high co-worker support, mental health training, and household income were significant predictors of MHSU. To further improve the study, I would suggest the authors to incorporate the following edits:

1. Page 3 Introduction Section: Its hard to understand which percentage refers to which mental health condition in this sentence. Please consider restructuring this sentence to “The systematic review of mental health among HCWs managing the COVID-19 pandemic revealed varying prevalence rates: anxiety had the highest rate at 67.6%, followed by stress at 62.9% and depression at 55.9%. While the lowest rates were observed for depression at 12.1%, stress at 29.8%, and anxiety at 24.1%”.

2. Page 4, Introduction Section: While its good that the authors have mentioned the ABMHSU in the introduction section, I would recommend shifting the details about the factors influencing this model to the methods section. By doing so the content will be streamlined, making it easy to understand and avoiding any unnecessary details in the introduction.

3. Page 4-5, Introduction Section: The paragraph discussing the impact of low MHSU contains a lot of repetitive information about the negative consequences experienced by HCWs. Hence, I would recommend the authors to condense these points to improve the clarity and focus of the text.

4. Methodology Section: Please consider briefly elaborating on what the Cronbach's alpha and Cohen's kappa values represent and what their importance is in the study. This will help readers appreciate the methodology of the study.

5. Page 18, Discussion Section: The sentence, "The high prevalence in our study was attributed to the utilisation of services for screening purposes which accounted for 96.3%."is ambiguous. Please explicitly state whether the 96.3% refers to the percentage of total MHSU or any other measure to ensure clarity.

6. Discussion Section: To improve the organization of the discussion section I would recommend the authors to divide it into further subsections like “impact of mental health training”, “socioeconomic factors” “supervisor support,” etc. This will make it easier for the readers to understand and navigate through the topics discussed.

6. PLOS authors have the option to publish the peer review history of their article (what does this mean?). If published, this will include your full peer review and any attached files.

Reviewer #1: No

Reviewer #2: No

---

## [Author Response · Author response to Decision Letter 0]

9 Sep 2024

Dear Editors,

Thank you for the opportunity to revise our manuscript. We have carefully addressed all the comments and suggestions provided by the reviewers. Please find our detailed responses to each comment in the attached letter response to reviewers.

We hope that our revisions meet your expectations and look forward to your feedback.

Best regards,

Corresponding Author

---

## [Decision Letter · Decision Letter 1]

3 Jan 2025

The drivers of mental health service utilisation among public primary healthcare workers in Negeri Sembilan

PONE-D-24-20993R1

Dear Dr. Ariffin,

We’re pleased to inform you that your manuscript has been judged scientifically suitable for publication and will be formally accepted for publication once it meets all outstanding technical requirements.

Kind regards,

Tariq Jamal Siddiqi

Academic Editor

PLOS ONE

Additional Editor Comments (optional):

Reviewers' comments:

Reviewer's Responses to Questions

**Comments to the Author**

1. If the authors have adequately addressed your comments raised in a previous round of review and you feel that this manuscript is now acceptable for publication, you may indicate that here to bypass the “Comments to the Author” section, enter your conflict of interest statement in the “Confidential to Editor” section, and submit your "Accept" recommendation.

Reviewer #3: (No Response)

Reviewer #4: (No Response)

2. Is the manuscript technically sound, and do the data support the conclusions?

Reviewer #3: Yes

Reviewer #4: Yes

3. Has the statistical analysis been performed appropriately and rigorously? 

Reviewer #3: Yes

Reviewer #4: Yes

4. Have the authors made all data underlying the findings in their manuscript fully available?

Reviewer #3: Yes

Reviewer #4: Yes

5. Is the manuscript presented in an intelligible fashion and written in standard English?

Reviewer #3: Yes

Reviewer #4: Yes

6. Review Comments to the Author

7. PLOS authors have the option to publish the peer review history of their article (what does this mean?). If published, this will include your full peer review and any attached files.

Reviewer #3: **Yes: **Akash Kumar

Reviewer #4: **Yes: **Ahmed Kamal Siddiqi

---

## [Editor Report · Acceptance letter]

12 Jan 2025

PONE-D-24-20993R1 

PLOS ONE

Dear Dr. Ariffin, 

I'm pleased to inform you that your manuscript has been deemed suitable for publication in PLOS ONE. Congratulations! Your manuscript is now being handed over to our production team.

Kind regards, 

on behalf of

Dr. Tariq Jamal Siddiqi 

Academic Editor

PLOS ONE